# Gigastep - One Billion Steps per Second Multi-agent Reinforcement Learning

**Mathias Lechner**
MIT CSAIL
mlechner@mit.edu

**Lianhao Yin**
MIT CSAIL
lianhao@mit.edu

**Tim Seyde**
MIT CSAIL
tseyde@mit.edu

**Tsun-Hsuan Wang**
MIT CSAIL
tsunw@mit.edu

**Wei Xiao**
MIT CSAIL
weixy@mit.edu

**Ramin Hasani**
MIT CSAIL
rhasani@mit.edu

**Joshua Rountree**
United States Air Force
rountree@mit.edu

**Daniela Rus**
MIT CSAIL
rus@csail.mit.edu

## Abstract

Multi-agent reinforcement learning (MARL) research is faced with a trade-off: it either uses complex environments requiring large compute resources, which makes it inaccessible to researchers with limited resources, or relies on simpler dynamics for faster execution, which makes the transferability of the results to more realistic tasks challenging. Motivated by these challenges, we present Gigastep, a fully vectorizable, MARL environment implemented in JAX, capable of executing up to one billion environment steps per second on consumer-grade hardware. Its design allows for comprehensive MARL experimentation, including a complex, high-dimensional space defined by 3D dynamics, stochasticity, and partial observations. Gigastep supports both collaborative and adversarial tasks, continuous and discrete action spaces, and provides RGB image and feature vector observations, allowing the evaluation of a wide range of MARL algorithms. We validate Gigastep's usability through an extensive set of experiments, underscoring its role in widening participation and promoting inclusivity in the MARL research community. MIT licensed code is available at https://github.com/mlech26l/gigastep.

## 1 Introduction

Decentralized partially observable Markov decision processes (Dec-POMDPs) [4, 42] represent a practically relevant yet largely unsolved challenge within the domain of deep reinforcement learning (Deep RL). These decision-making frameworks involve multiple agents operating in an environment where each agent has access to only partial observations, making it necessary to reason about both the global state and the private information of individual agents. Solving Dec-POMDPs efficiently and effectively has profound implications for real-world applications [9], such as human-machine interaction [30], multi-robot systems [13], autonomous driving [28, 29, 10], and decentralized control of complex networks [71, 11].

To address this challenge, numerous environments have been proposed, each with unique strengths and weaknesses. For instance, the Starcraft Multi-Agent Challenge (SMAC) [50] emerged as a popular platform for benchmarking Dec-POMDP algorithms. However, the SMAC v2 [15] revealed that this environment is fundamentally deterministic, allowing agents to memorize action sequences rather than developing genuinely generalizable behavior, thus providing a skewed perspective on algorithmic performance. Consequently, it is essential to question the robustness and effectiveness of the solutions that have been evaluated only in single environments.

37th Conference on Neural Information Processing Systems (NeurIPS 2023) Track on Datasets and Benchmarks.

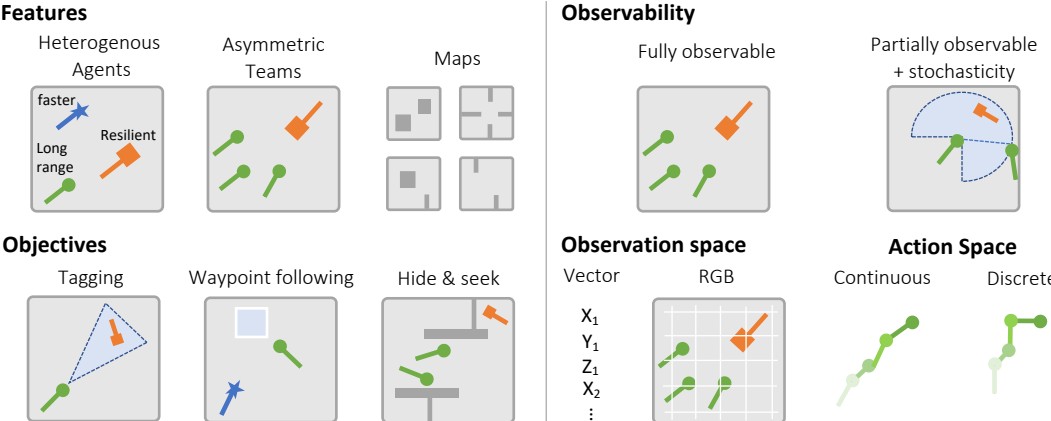

Figure 1: An overview of Gigastep's features and characteristics

Furthermore, the current landscape of Dec-POMDP research exhibits a stark dichotomy: on the one hand, we have environments that, although computationally accessible, lack the complexity to adequately represent the intricacies of real-world Dec-POMDP scenarios. These are often overly simplistic, two-dimensional, and void of the partial observability that characterizes Dec-POMDPs [37]. On the other hand, we have environments that are sufficiently complex but require such a high degree of resources that exclude many researchers, particularly those from underrepresented or underfunded institutions, thus hindering the democratization and inclusive progress of the field [55].

Motivated by these concerns, our paper aims to bridge this gap by introducing a novel benchmark environment called Giagstep for Dec-POMDPs that balances difficulty, realism, and computational accessibility. By addressing the identified limitations of existing methods, we hope to foster a more inclusive, comprehensive, and realistic exploration of Dec-POMDPs in deep reinforcement learning research. In Gigastep, a set of airborne agents separated into two teams navigate in the 3-dimensional space and need to achieve predefined objectives, such as following waypoints and tagging, chasing, or avoiding agents of the other team. Each agent perceives their neighborhood and shares information with allied teammates stochastically, making it an ideal instance of benchmark Dec-POMDPs. We summarize our contributions in three points:

- Gigastep is designed with efficiency in mind enabling MARL on consumer-grade hardware
- Gigastep is fully open source allowing customization, specialization, and further advancements by the research community
- Gigastep captures the open challenges in multi-agent reinforcement learning in decentralized partially observable Markov decision processes (Dec-POMDP)

## 2 Related Works

In this section, we list related RL and MARL benchmark environments and describe their differences from Gigastep. Moreover, in Table 1, we summarize key features and lack thereof of the environments.

A wide range of simulators are designed to help develop reinforcement learning algorithms to solve problems that can be treated as Dec-POMDPs or general MDPs. These simulators focus on different aspects such as multi-player computer games [49, 15], sports [23], social interactions [34], life evaluations [60], particle simulation [64, 73], differentiable rigid body dynamics [17, 69], drones [54, 58, 43, 18, 65], etc. We elaborate on their advantages and limitations in the following paragraphs.

**Drones simulators** – Multiple drone simulators are developed for different research questions [54, 18, 58, 43]. Some of them focus on photo-realistic simulation such as Airsim [54], flight goggles [18], flightmare [58], etc. Some of them aim to simulate multi-drones for reinforcement learning such as Pybullet-drones [43] and Airsim [54]. However, they are not designed to be vectorized and run on GPU. The speed slows down significantly when the number of drones increases.

**StarCraft Multi-Agent Challenge** — The StarCraft Multi-Agent Challenge (SMAC) [49] and SMACv2 [15], are of most prominent MARL benchmarks. SMAC offers a complex, real-time, and partially observable platform for MARL research based on the popular game StarCraft II.

However, there are certain limitations to the SMAC environment that Gigastep aims to address. Firstly, while SMAC does offer multiple units and scenarios, the game-based nature inherently limits the variability and control over the environment's parameters. Our environment is fully open source and designed to be more flexible, enabling researchers to customize and design scenarios that match their specific research goals. Secondly, SMAC's computational efficiency is tied to the StarCraft II game engine, which can be a bottleneck for large-scale experiments and severely limits its access to underrepresented researchers with little compute infrastructure. Our JAX-based environment is designed from the ground up to exploit hardware accelerators like GPUs and TPUs, ensuring highly efficient simulations. Lastly, SMAC's interface to the environment, through PySC2, can be cumbersome to navigate for newcomers to the field. Our environment is engineered to be user-friendly, with an intuitive API that allows easy integration with various reinforcement learning algorithms and custom scenarios.

**PettingZoo** — PettingZoo is a shared Python interface for a range of multi-agent reinforcement learning (MARL) challenges. It includes an extensive collection of benchmark environments such as the multi-agent Arcade learning environment (ALE) [3], the Multi Particle Environments (MPE) [36], SISL (Stanford Intelligent Systems Laboratory) environments [19] as well as a few of other environments. While PettingZoo [64] provides a valuable set of MARL environments and tools, its environments include a few limitations that Gigastap aims to resolve. For instance, the multi-agent ALE environments comprise of only 1vs1 player games, whereas Gigastep allows teams of more than a thousand agents to play simultaneously. Moreover, compared to Gigastep, the MPE and the SISL environments possess only basic 2D dynamics, lack support for high-dimensional RGB image observations, and contain at most 8 agents.

**brax and gymnax** — Brax [17] and Gymnax [25] are single-agent RL environments that are entirely implemented in JAX. Consequently, these environments can fully leverage the computational power of GPU and other accelerator devices to gather experience data for RL training in parallel, enabling fast RL experimentation. However, they lack native support for multi-agent settings. Gigastep instead aims to bridge this gap and replicate the advantages of both Brax and Gymnax in multi-agent settings.

**Magent** – Magent [73] is a research platform for multi-agent reinforcement learning. It has a specific aim to observe and understand the social behaviour of agents and individual behaviour through interaction and communications. However, it only provides adversarial scenarios such as pursuit, gathering, and battle. It has no built-in vectorization and can only be parallel executed by creating workers, which is not computationally efficient for state-of-the-art reinforcement learning algorithms.

**Google football** – Google football [23] is design to simulate football game in particular. It is limited to the number of team members in the cooperative and competitive teams which represents a sports game. Therefore, the scalability in both the number of agents and scenarios is limited.

**Melting Pot** – Melting Pot [34] has the capability to evaluate the generalization of algorithms to novel social situations such as cooperation, competition, deception, reciprocation, trust, stubbornness, and so on. However, it is limited in the number of agents and can not be accelerated in GPU.

**Multi-agent Mujoco** – Multi-agent Mujoco [44] is a research platform that treats each joint of a robot as an individual agent, upon which, one can develop multi-agent algorithms to control a full robot. However, there is no higher level and task-aware cooperation and competition between multi-agents.

**Multi-agent RL algorithms** – Effective MARL algorithms need to scale across high-dimensional action spaces and enable coordination among distributed agents. Independent learners [61] provide scalability at the cost of exacerbating coordination challenges, particularly under partial observability [12, 39]. Fully centralized learners can become computationally infeasible in high-dimensional action spaces, and distributed methods may improve learning [51, 47]. A common approach within Q-learning is the decomposition of the joint value function into local utilities [45, 57, 63, 53], which similarly transfers to the actor-critic setting [66, 59, 44] and hierarchical control [67]. More generally, the concept of centralized training with decentralized execution enables effective training of distributed teams under partial observability. Interactions can further be limited to only local interplay among agents [70, 5]. Alternatively, evolutionary approaches side-step the need for dense gradient information and may improve coordination under sparse team-centric reward feedback [38, 62].

Table 1: Comparison of different RL and multi-agent RL environments

| Environment | Collaborative multi-agent | Adversarial multi-agent | Partial observability | Stochastic dynamics | Heterogeneous agents | > 1000 agents | Open source | Vector observations | RGB image observations | 3D dynamics | Built-in vectorization | GPU acceleration | Customize environment |
|---|---|---|---|---|---|---|---|---|---|---|---|---|---|
| gym [8] | ✓ | ✓ | ✓ | ✓ | - | - | ✓ | ✓ | ✓ | ✓ | ✓ | - | ✓ |
| gymnax [25] | - | - | - | ✓ | - | - | ✓ | ✓ | ✓ | - | ✓ | ✓ | - |
| brax [17] | - | - | - | - | - | - | ✓ | ✓ | - | ✓ | ✓ | ✓ | ✓ |
| SMAC [49] | ✓ | ✓ | ✓ | - | ✓ | - | ✓ | ✓ | - | - | - | - | - |
| SMACv2 [15] | ✓ | ✓ | ✓ | - | ✓ | - | ✓ | ✓ | - | - | - | - | - |
| ALE [3] | - | ✓ | - | - | - | - | ✓ | ✓ | ✓ | - | - | - | - |
| Minecraft MALMO [21] | - | - | ✓ | - | ✓ | ✓ | - | - | ✓ | ✓ | - | - | - |
| Deepmind Lab [2] | - | - | ✓ | - | - | - | ✓ | ✓ | ✓ | - | - | - | ✓ |
| FortAttack [14] | ✓ | ✓ | - | - | ✓ | - | ✓ | ✓ | - | - | - | - | - |
| Derk's Gym [41] | ✓ | ✓ | - | - | - | - | - | ✓ | - | - | - | ✓ | - |
| Hanabi [1] | ✓ | - | ✓ | ✓ | - | - | ✓ | ✓ | - | N/A | - | - | - |
| Neural MMO [60] | ✓ | ✓ | ✓ | - | ✓ | ✓ | ✓ | - | ✓ | - | - | - | ✓ |
| MPE in PettingZoo [64] | ✓ | ✓ | ✓ | - | - | - | ✓ | ✓ | ✓ | - | - | - | - |
| SISL in PettingZoo[19] | ✓ | ✓ | - | - | ✓ | - | ✓ | ✓ | ✓ | - | - | - | - |
| jumanji [6] | ✓ | ✓ | ✓ | ✓ | - | - | ✓ | ✓ | ✓ | - | ✓ | ✓ | - |
| Google football [23] | ✓ | ✓ | - | ✓ | - | - | ✓ | ✓ | ✓ | ✓ | - | ✓ | - |
| Multi-agent Mujoco [44] | ✓ | - | - | - | - | - | ✓ | ✓ | - | ✓ | - | - | - |
| Pomerman [46] | ✓ | ✓ | ✓ | - | ✓ | - | ✓ | ✓ | ✓ | - | - | - | - |
| Melting Pot [34] | ✓ | ✓ | ✓ | ✓ | ✓ | - | ✓ | - | ✓ | - | - | - | - |
| Magent [73] | ✓ | ✓ | - | - | ✓ | ✓ | ✓ | ✓ | - | - | ✓ | ✓ | ✓ |
| Gym-pybullet-drones [43] | ✓ | - | - | - | - | - | ✓ | ✓ | ✓ | ✓ | ✓ | ✓ | ✓ |
| Air-sim [54] | ✓ | - | - | - | - | - | ✓ | ✓ | ✓ | ✓ | - | - | - |
| Flightmare [58] | ✓ | - | - | - | - | - | ✓ | ✓ | ✓ | ✓ | ✓ | - | - |
| **gigastep (ours)** | ✓ | ✓ | ✓ | ✓ | ✓ | ✓ | ✓ | ✓ | ✓ | ✓ | ✓ | ✓ | ✓ |

## 3  Designing Gigastep

In this section, we first provide a formal definition of Dec-POMDPs, followed by a detailed description of the design choices made with Gigastep.

### 3.1  Preliminaries

A Decentralized Partially Observable Markov Decision Process (Dec-POMDP) is an extension of a Markov Decision Process (MDP) that facilitates the modeling of multi-agent systems within partially observable environments. Specifically, a Dec-POMDP introduces decentralization, in which each agent makes decisions based on its individual actions and observations without direct communication or knowledge of other agents' observations or actions.

Formally, a Dec-POMDP is defined by a tuple $G = (n, S, A, T, \Omega, O, R, \gamma)$ where: $n$ is the number of agents. $S$ is a finite set of states of the environment. $A = \{A_1, A_2, \ldots A_n\}$ is a set of action sets, where $A_i$ is the set of actions available to agent $i$. The combined action space of all agents is $A = A_1 \times A_2 \times \ldots \times A_n$. $T : S \times A \to \Delta(S)$ is the state transition function, where $\Delta(X)$ denotes a probability distribution of the set $X$. For a given state $s \in S$ and joint action $a \in A$, $T(s, a)$ is the probability distribution representing probability $\mathbb{P}(s'|a, s)$ of transitioning to state $s'$ from state $s$ when joint action $a$ is taken. $\Omega = \{\Omega_1, \Omega_2, \ldots, \Omega_n\}$ is a set of observations, where $\Omega_i$ is the set of observations for agent $i$. The combined observation space of all agents is $\Omega = \Omega_1 \times \Omega_2 \times \ldots \times \Omega_n$.

| Maps + Waypoints | Stochastic observation cone | 1000 vs 1000 agents | Heterogenous agent types |
|---|---|---|---|

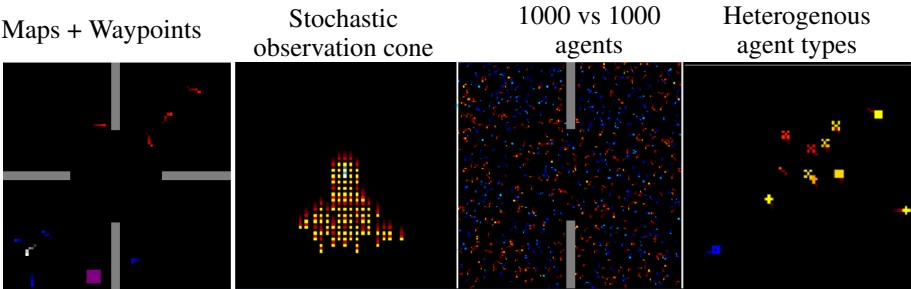

Figure 2: Examples of the RGB image observation mode.

$O : S \times A \to \Delta(\Omega)$ is the observation function. For a given joint action $a \in A$ and state $s$, $O(s, a)$ is the probability $\mathbb{P}(o|a, s)$ that agents observe $o$ after taking joint action $a$ and in state $s$. $R : S \times A \to R$ is the reward function with $R(s, a)$ representing the immediate reward for taking joint action $a$ in state $s$. $\gamma \in [0, 1]$ is the discount factor, which determines the present value of future rewards.

Each agent $i \in \{1, 2, \ldots n\}$ selects actions based on its policy $\pi_i : o_i \to \Delta(A_i)$, which is a mapping from its observation to a distribution over its action set $A_i$. The joint policy of all agents is then $\pi = (\pi_1, \pi_2, \ldots, \pi_n)$. The goal in a Dec-POMDP is to find an optimal joint policy $\pi^*$ that maximizes the expected discounted sum of rewards $\mathbb{E}(\sum_{t=0}^{\infty} \gamma^t R(s_t, a_t)|\pi, s_0 = s)$, where $s_t$ and $a_t$ are the states and joint action at time $t$, respectively.

### 3.2 Design considerations

Here, we discuss the design choices and constraints made in Gigastep.

**Vectorizable environment stepping** RL algorithms learn from their experience of interacting with the environment. Consequently, their training is limited by the amount of experience data that is available. However, generating this data can be a slow process, especially in complex environments where each interaction could take a considerable amount of computing time. This bottleneck significantly impacts the efficiency of experimentation with RL algorithms and hinders research progress. The limitation is further amplified for underrepresented researchers who may lack the funding for vast computer resources.

Therefore, there is a pressing need for a fast and vectorizable environment that can handle environment rollouts in parallel. Such an environment would enable mult-agent reinforcement learning algorithms to generate and learn from a larger amount of experience data in a shorter span of time, thereby significantly accelerating research progress. A vectorizable environment would also allow these algorithms to fully exploit the capabilities of modern GPUs, further enhancing their computational efficiency and speed.

We resolved this issue by implementing Gigastep fully in JAX [7], an accelerated linear algebra (XLA) package that allows compiled programs to run in parallel on CPU or GPU devices. Moreover, compared to existing approaches Gigastep support rending the agents observations as RGB images in a fully vectorized and `jax.jit` compilable fashion. Contrarily, existing benchmarks often provide only vectorized feature vector observations and rely on third-party game libraries for rending the agents' state and observation.

**Collaborative and adversarial components** An ideal MARL benchmark integrates both collaborative and adversarial components within the environment. The simultaneous existence of these aspects within an environment enables a more comprehensive emulation of complex real-world scenarios, thereby fostering enhanced learning capabilities in agents. The collaborative component encourages cooperative strategies, enhancing the ability of agents to work together towards common goals, while the adversarial aspect fosters competitiveness, which hones an agent's ability to adapt to an ever-changing environment and devise individualistic strategies. Together, these components not only enrich the learning experience but also improve the robustness of the agent's decision-making process, fostering a more nuanced understanding of their environment, consequently leading to improved performance and adaptability in diverse and dynamic contexts.

**Stochastic dynamics** In MARL, it is crucial to design environments that possess a level of stochasticity in their dynamics. This element of randomness is important to prevent agents from learning and exploiting deterministic action sequences, thereby encouraging more robust and adaptable learning. Moreover, stochasticity is a fundamental aspect in many real-world applications [32, 33], thus critical if MARL algorithms need to transfer from simulation to real-world deployment. The Starcraft Multi-Agent Challenge (SMAC) v1 is a prime example of a deterministic environment where agents were able to learn simple, reproducible action patterns that led to an overestimation of their abilities [49, 15]. The introduction of stochasticity in SMAC v2 [15], has considerably improved the testing of learned policies by introducing uncertainty and forcing agents to devise strategies that are resilient to changing conditions. Hence, incorporating stochasticity into MARL environments is a necessary step toward achieving more comprehensive and realistic evaluations of agent competencies.

### 3.3 Environment description

In this section, we discuss various features and characteristics of Gigastep. An overview of these features is illustrated in Fig. 1. In particular, Gigastep was inspired by multiplayer online battle arena (MOBA) style games, where two teams of agents compete with each other. The agents have heterogeneous capabilities, which enables the emergence of complex team dynamics. Complexity is often further added by partial observability of the game state and stochastic or low-bandwidth communication with a team.

**Agents' features.** The environment features agents that move in 3D rectangular landscapes. The agents are heterogeneous in their skill sets. For instance, agents could be faster, more resilient, and possess a longer field of view. The environment allows for symmetric and asymmetric team configurations where one team has more agents than the opponent.

**Maps.** Gigastep comes with customize maps in both 2D and 3D.

**Objectives.** The agents are split into two teams with configurable objectives such as tagging, chasing waypoints, and hide & seek. For the tagging objective, the goal revolves around tagging opponents to eliminate them, achieving victory when all enemy agents are tagged. For the chasing waypoint objective, random waypoints appear on the map, and points are awarded to the team whose agent reaches the waypoint first. Notably, Gigastep implements a physicality to the environment, where agents can collide with each other or the map boundaries, leading to their elimination for the rest of the episode. For the hide-and-seek objective, agents of team one attempt to catch agents of team two. Team one agents get rewards when they catch opponent agents, while team two agents are rewarded for staying alive the longest.

**Observability.** Moreover, Gigastep offers both full and partial observability (PO) schemes. In the PO setting, agents have a *visibility* cone within which they stochastically perceive opponent agents, with increasing detection probability the closer the foes are. Moreover, Gigastep incorporates a stochastic communication system whereby allied agents share information about detected enemies based on distance-related probability. This complex environment fosters intricate multi-agent strategies, providing a rich testbed for RL methodologies.

**Action spaces.** The action space of each agent consists of three dimensions: heading, acceleration, and climbing. With the heading dimension, the agent can change its movement direction on the xy-plane. With acceleration, the agent controls its velocity. Finally, with the climbing dimension, the agent can move along the z-axis in the physical space.

Gigastep provides both continuous and discrete action spaces. In the continuous setting, each action is clamped to the interval $[-1, 1]$. For the discrete action space setting we discretized the actions to $\{-1, 0, +1\}^3$, i.e., 27 possible actions.

**Observation spaces.** Gigastep supports two formats for presenting observations: feature vectors and RGB images. The feature vector format constructs a fixed-length vector of numerical features that represent the agent's own 3D state as well as the state of the $k$ nearest other agents that are seen by the ego agent. The state data contains variables such as the location, heading angle, agent type, and health status. The feature vector observation format allows the straightforward use of multi-layer perceptron (MLP) or recurrent neural network (RNN) policies [31, 20, 27].

The RGB image observation format renders the agent's 3D view of the environment as a 2-dimensional array of pixels, each containing three color channels of 8-bit resolution. To render the $x$,$y$, and $z$

Table 2: Selection of the scenarios included in Gigastep. The objectives are waypoint following (WF), hide and seek (HF), and tagging (TA). Build-in agent types are D (default), H (increased health), F (faster speed), R (increased beam range), and S (all stats increased).

| Scenario name and Objective | | Description |
| --- | --- | --- |
| `waypoint_3_vs_3` | WF | |
| `waypoint_5_vs_3` | WF | Asymmetric teams (5×D vs 3×F) |
| `hide_and_seek_5_vs_5` | HS | |
| `hide_and_seek_5_vs_10` | HS | Asymmetric teams (5×D vs 10×D) |
| `identical_1_vs_1` | TA | |
| `identical_5_vs_5` | TA | |
| `identical_5_vs_3` | TA | Asymmetric teams (5×D vs 3×S) |
| `special_5_vs_5` | TA | Heterogeneous agents $1 \times \{D, H, F, R, S\}$ per team |
| `special_20_vs_20` | TA | Heterogeneous agents $5 \times \{D, H, F, R\}$ per team |
| ... | | ... |

location, the heading, altitude, and agent type in the RGB image, we took inspiration from the representation of the traffic collision avoidance system (TCAS) [22] which is used in commercial avionics to visualize multiple aircraft on a small screen, e.g., see Figure 2 for example renderings of Gigastep. In essence, TCAS uses color and shape to visualize three or more dimensional features on a two-dimensional image. We chose a default resolution of 84-by-84 pixels to be compatible with the resolution typically used with the arcade learning environment's (ALE)[3, 40] and many RL frameworks such as rllib [35]. Unlike existing MARL benchmarks, the rendering process is fully implemented in JAX and can be compiled using `jax.jit` and vectorized using `jax.vmap`.

**Scenarios** An ideal MARL benchmark must provide a high degree of flexibility, enabling users to construct custom scenarios that encompass a diverse range of configurations. This ranges from simple one vs one setups to expansive environments involving thousands of potentially heterogeneous agents. Gigastep, embraces this principle by offering a flexible platform for custom scenario generation. It allows users to craft unique maps, define custom agent types, build teams of size one to over a thousand, and specify multiple objectives, among other customizations. This robust, user-oriented framework encourages exploration across the vast design space inherent to MARL problems, promoting creativity, adaptability, and thoroughness in the pursuit of MARL advancements.

## 4 Environment Performance

In this section, we run a performance evaluation to test the rollout speed of Gigastep. In particular, we run the environment in a closed loop with a neural network policy using `jax.jit` and `jax.vmap` while increasing the batch size of the vmap dimension. We repeat this experiment on several commonly used consumer GPUs and for both observation space types, i.e., feature vector and RGB image. The results are shown in Figure 3. They demonstrate that a sufficient amount of experience data can be generated with Gigastep on affordable hardware within a short runtime.

## 5 MARL Baselines with Gigastep

To evaluate the developed environment for multi-agent reinforcement learning research, we adopted a handful of MARL algorithms such as multi-agent evolutionary strategies and multi-agent reinforcement learning algorithms. We discovered the emergence of a diverse set of behaviors using the proposed methods, validating the suitability of Gigastep for MARL research. We want to highlight that our objective is to validate Gigastep and not to run a comprehensive comparison of different MARL algorithms. Additional details and results of multiple MARL baselines can be found in the supplementary materials.

Self-play has shown to be the most effective method to train agents in adversarial settings [56]. We, therefore, adopted self-play in the following algorithms, but keep the policy parameters of both teams separated to allow asymmetric team structures.

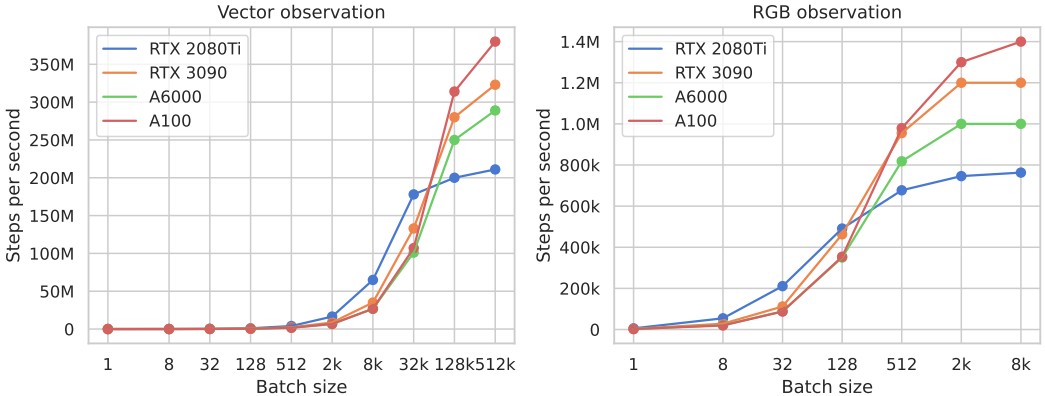

Figure 3: Performance of Gigastep in a 2-vs-2 agent scenario. Number of agent steps that can be computed within a second.

**Multi-agent PPO.** We tested multi-agent proximal policy optimization (PPO) [72] using the standard PPO implementation [52] with centralized value function $V_\phi(S) = f(V_{\phi_i}(s_i)_{i=1,...,n})$ and decentralized action policy $a_i = \pi_i(s_i)$. During training, the value function was centralized and trained with the composition of the individual value functions, where each individual value function $V_{\phi_i}$ uses local observation from individual agent $s_i$. During the execution phase, the action policy $\pi_i(s_i)$ is fully decentralized. The individual value function $V_{\phi_i}$ and $\pi_i$ share the same structure and parameters among each agent. In Figure 4 we visualize example behaviors that emerged during the MARL training procedure on some Gigastep environments.

**Multi-agent evolution strategy.** We interfaced Gigastep with evosax [24] to facilitate streamlined training of various evolution strategies. This formulation enables scanning through entire episodes, further reducing communication overhead and runtime requirements by only updating based on episode returns. We specifically evaluate multi-team OpenAI-ES [48], separable NES [68], and Discovered Evolution Strategy [26], which all successfully learn diverse behaviors in Gigastep.

## 6  Benchmark Challenges on top of Gigastep

Evaluation is one of the issues of MARL for Dec-POMDP, especially in an adversarial setting and self-play. Specifically, reward is only defined relative with respect to a particular adversary policy, i.e., an old copy of the policy in case of self-play, but not an absolute measure as found in single-agent RL.

MOBA or board games such as chess solve this issue by using an Elo-based [16] ranking system, which defines a numerical skill ranking over a set of players or policies.

A key assumption required to adapt such an Elo-ranking for Gigastep is a large pool of fixed policies that reflect a wide range of skill levels. To obtain this set, we define manually coded and MARL-trained baseline policies for Gigastep. The manually coded policies implement only elementary behaviors, such as a team moving in circles. We observed that such simple behavior already poses a decent baseline due to two properties: First, the circling behavior strictly outperforms random policies due to random agents eventually crashing at the boundary of the environment while the circling team moves on a stable orbit indefinitely. Second, for tracking objectives, the view of a circling team can cover a large part of the maps and simultaneously implement some rudimentary evasion strategy. Consequently, the performance of a policy against our manually coded baseline allows us to measure if the policy has learned the objective of the task beyond random behavior. For instance, in Figure 5a, we show the progress of the win rate of multi-agent PPO-trained policies against the manually coded baselines. At the beginning of the training, we see that the manually coded agents strictly outperform the RL agents at an 80:20 win-loose ratio. Nonetheless, after a certain amount of training iterations, the RL agents are able to win nearly all engagements with the coded baselines.

Our second baseline policy set is neural networks trained via multi-agent PPO and self-play. To obtain policies across diverse skill levels, we save checkpoints throughout the training process, resulting in

**A)** Line formation

**B)** Climbing (gaining potential energy)

**C)** Encircling

**D)** Grouping

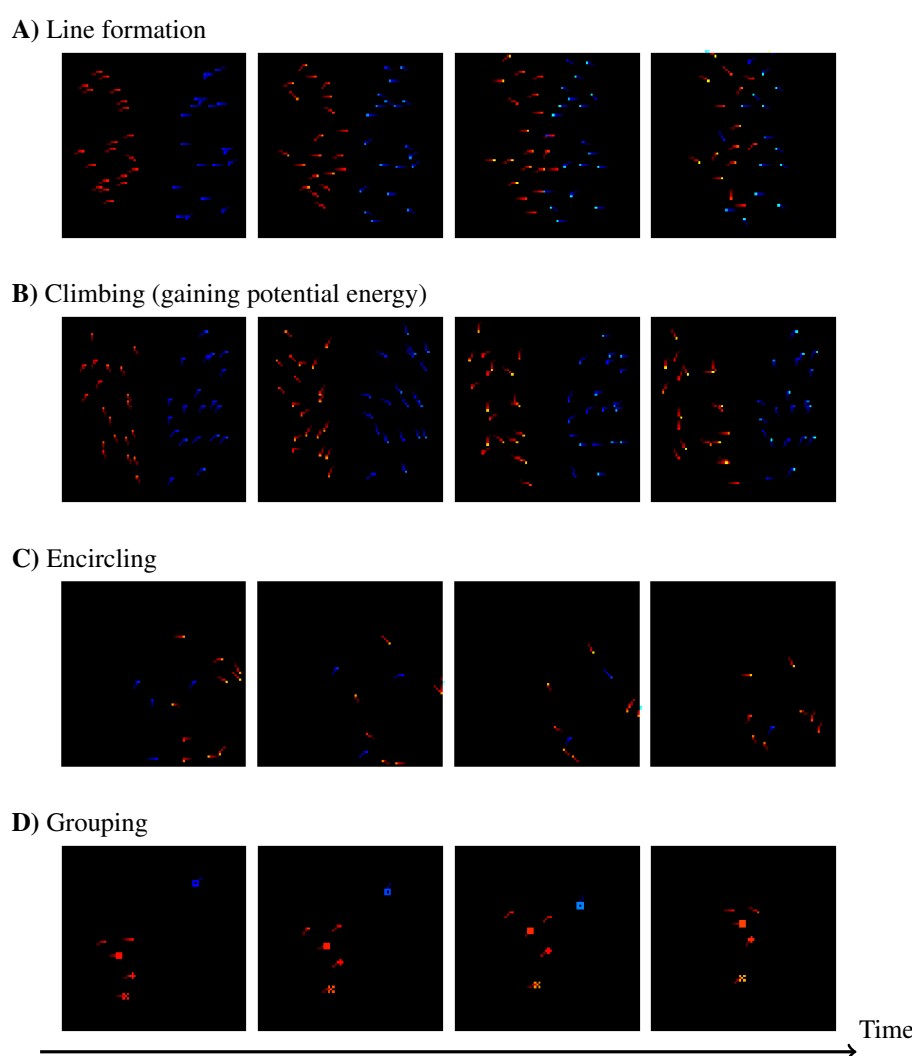

Time

Figure 4: Visualization of some examples of the team behavior emerging during the MARL training process on Gigastep environments.

weaker baselines at the beginning of the training and stronger ones toward the end. An example of a MARL training progress using an Elo-ranking system is shown in Figure 5b.

In the future, we may extend the pool of baseline policies in case advances in multi-agent RL algorithms are able to train much stronger policies that render the current pool inaccurate.

## 7   Limitation

We have provided demonstrative algorithms to train agents for the Gigastep that show the emergence of competitive agents. However, we did not perform extensive tuning for all game configurations. Consequently, we may update certain scenarios if necessary, e.g., a scenario that turns out to be too simplistic or difficult to allow the emergence of complex agent behaviors.

Additionally, we want to distinguish between Gigastep as a research platform to create scalable and complex MARL environments and the built-in benchmark challenge defined in Section 6. The flexibility and openness of Gigastep allow future research to create additional challenges with Gigastep, including custom scenarios, other objectives, and different agent capabilities and dynamics.

Finally, while Gigastep provides a wide range of objectives and other environmental parameters, we highlight that all evaluations with Gigastep share common agent dynamics. This can potentially

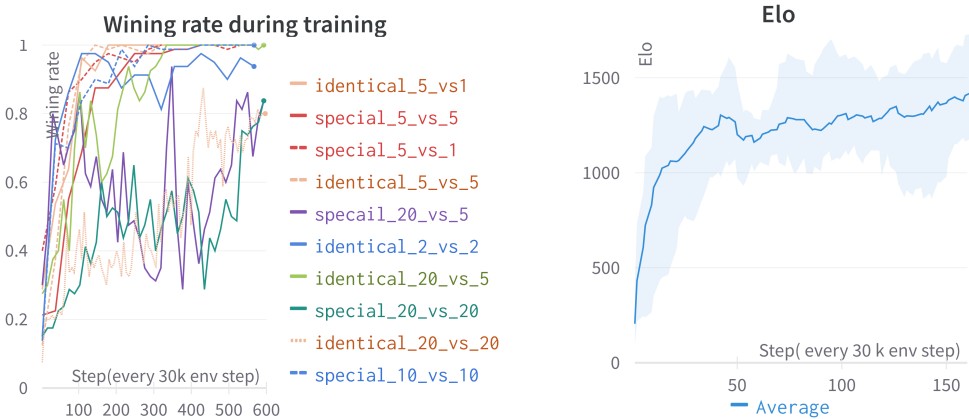

(a) Win rate of a policy trained via multi-agent PPO against a set of predefined baselines.

(b) Evaluation of a policy trained via multi-agent PPO using an Elo-based rating system in Gigastep.

Figure 5: Agent performance over the course of a MARL learning process with Gigastep.

introduce bias into MARL experimentation if Giagstep is used as the sole evaluation platform. Consequently, we note that proper MARL experimentation requires a diverse set of environments for the robustness and transferability of the observed results.

# 8 Conclusion

In this work, we introduced Gigastep, a novel, fully vectorizable multi-agent reinforcement learning (MARL) environment designed in JAX. Gigastep distinguishes itself by its ability to carry out up to one billion steps per second on consumer-grade hardware, a feature that significantly broadens the accessibility of MARL research. This environment encompasses complex, high-dimensional domains, integrating 3D dynamics, stochasticity, and partial observability. It also supports both collaborative and adversarial tasks, making it a comprehensive platform for a wide range of MARL applications.

Our experiments demonstrate that Gigastep can play a pivotal role in the development of MARL algorithms with high efficiency. By enabling researchers to conduct advanced simulations on affordable hardware, Gigastep addresses a key challenge in the field - the computational divide that often impedes underrepresented researchers from engaging in cutting-edge MARL research.

As we continue to explore the potential of Gigastep, we anticipate that its capacity to democratize access to MARL research promote a more diverse, inclusive, and innovative scientific community. Future work will focus on extending Gigastep's capabilities, including the integration of more complex task domains and additional functionality to further enhance the user experience and expand the range of feasible MARL experiments. Furthermore, we aim to make use of Gigastep's RGB observation space capabilities for research on human-AI collaboration by adding user-controlled agents along RL agents. We hope that the introduction of Gigastep will inspire new directions in MARL research and contribute to the growth and diversification of our scientific community.

## Acknowledgments and Disclosure of Funding

Research was sponsored by the United States Air Force Research Laboratory and the United States Air Force Artificial Intelligence Accelerator and was accomplished under Cooperative Agreement Number FA8750-19-2-1000. The views and conclusions contained in this document are those of the authors and should not be interpreted as representing the official policies, either expressed or implied, of the United States Air Force or the U.S. Government. The U.S. Government is authorized to reproduce and distribute reprints for Government purposes notwithstanding any copyright notation herein. The research was also funded in part by the AI2050 program at Schmidt Futures (Grant G-22-63172) and Capgemini SE.

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

# A  Appendix

Gigastep is published under the open-source MIT license on Github [https://github.com/mlech26l/gigastep](https://github.com/mlech26l/gigastep). Further development and potential updates discussed in the Limitations section will take place on the Gigastep GitHub page.

**Accountability framework**

Our team is committed to maintaining an open and accountable framework throughout the development of Gigastep. Our work is being carried out with the utmost transparency and we welcome the participation and insights of the broader AI community. The primary objective of our project is to democratize access to efficient and affordable MARL experimentation. To this end, we ensure that every code commit, model modification, and development update is documented and shared publicly, to provide insights into the ongoing development process. We also uphold rigorous software testing standards and frequent internal code reviews to maintain code quality and reliability. Moreover, we actively encourage feedback, constructive criticism, and contribution from other researchers and developers, and have implemented mechanisms for easy reporting of issues and suggestions. By maintaining this level of accountability, we strive to build a resource that genuinely serves the needs of the MARL community and advances the field of AI research.

