# Appendix
# Gigastep - One Billion Steps per Second Multi-agent Reinforcement Learning

**Mathias Lechner**
MIT CSAIL
mlechner@mit.edu

**Lianhao Yin**
MIT CSAIL
lianhao@mit.edu

**Tim Seyde**
MIT CSAIL
tseyde@mit.edu

**Tsun-Hsuan Wang**
MIT CSAIL
tsunw@mit.edu

**Wei Xiao**
MIT CSAIL
weixy@mit.edu

**Ramin Hasani**
MIT CSAIL
rhasani@mit.edu

**Joshua Rountree**
United States Air Force
rountree@mit.edu

**Daniela Rus**
MIT CSAIL
rus@csail.mit.edu

## 1 MARL Baselines

In this section, we train policies for different scenarios to validate that the tasks defined in Gigastep can be solved with multi-agent RL algorithms. In particular, we use multi-agent PPO implemented in JAX.

In competitive or adversarial MARL, an objective reward measure is not defined, as the collected reward inherently depends on the relative strength of the opposing agent's policy. Therefore, to measure the training progress, we compare the current policy with previous checkpoints of the same policy at earlier training iterations. Specifically, an improving policy should be able to outperform its previous counterparts. For instance, for the identical 5 vs 5 scenario, in Figure 1a, we plot the win rate of a policy at different checkpoints (x-axis) compared to the same policy at every other checkpoint of the training process (y-axis). The plot shows a high win rate at the lower triangular part of the heatmap, demonstrating that the policy at later stages of the training is able to consistently win against earlier checkpoints. In Figure 1b, we plot the total accumulated win rate during training against all other checkpoints. Similarly, the curves show an increase in win rate during training and plateauing at later stages.

**Impact of the number of agents on the learning** We repeat this experiment for the same variant of the scenario with more agents. The results in Figure 1 (5 vs 5), Figure 2 (10 vs 10), and Figure 3 (20 vs 20) show that with an increasing number of agents, the training becomes less stable.

**Heterogenous agents** In Figure 4 (5 vs 5), Figure 5 (10 vs 10), and Figure 6 (20 vs 20) we plot the same experiment results for heterogeneous team compositions, showing a similar trend.

**Asymmetric team compositions** We also validate scenarios that have asymmetric teams, meaning that team A has a different number and types of agents than team B. Different from the experiments from before, here, we parameterize the policy of team A and team B with separate neural networks. For instance, in Figure 7, we show the win rates from the perspective of the team A policy. The major difference between the plots to the results of the symmetric team experiments is the absence of the triangular structure in the heatmap plot. This is caused by the asymmetry of the teams and the policy parametrization, i.e., the results were not obtained by playing against old checkpoints of itself but by a different policy that may consistently be weaker or stronger across longer periods of the training process. Moreover, the results show that due to the asymmetry, the environment favors one of the two

37th Conference on Neural Information Processing Systems (NeurIPS 2023) Track on Datasets and Benchmarks.

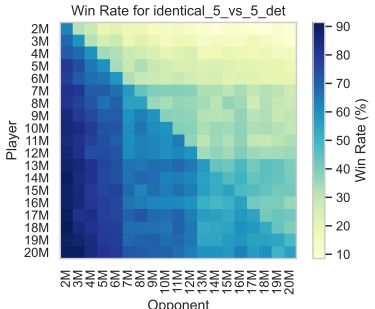

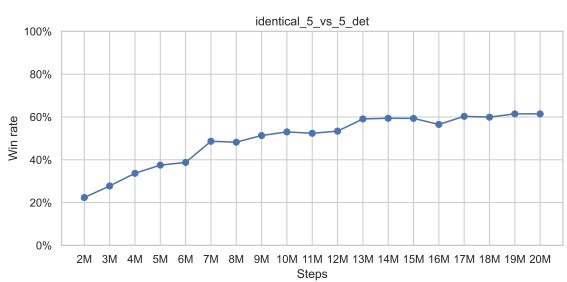

(a) Win rate when playing each policy check-point against each other checkpoint.

(b) Accumulated win rate of the different checkpoints (x-axis) with respect to all other checkpoints.

Figure 1: Learning progress of self-play with multi-agent PPO on the `identical_5_vs_5` scenario.

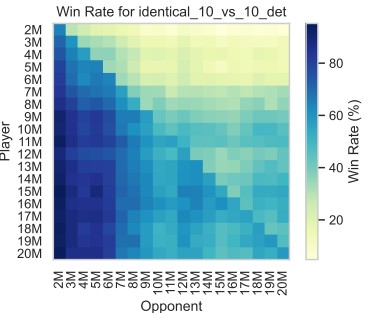

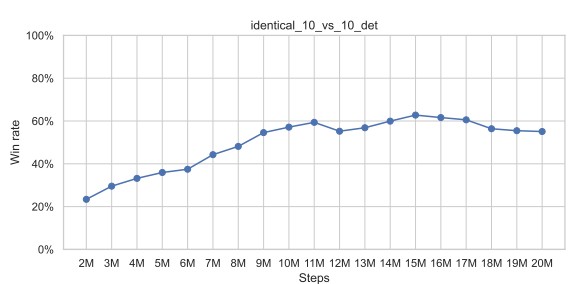

(a) Win rate when playing each policy check-point against each other checkpoint.

(b) Accumulated win rate of the different checkpoints (x-axis) with respect to all other checkpoints.

Figure 2: Learning progress of self-play with multi-agent PPO on the `identical_10_vs_10` scenario.

teams, especially at the beginning of the training process. In Figure 8 and Figure 9 we replicate the results for the asymmetric scenarios with 20 vs 5 agents and 10 vs 3 heterogeneous agents.

## 2 Analysis of learned agent behaviors

In this section, we describe the behaviors that emerged after the training process of some of Gigastep's scenarios. This analysis serves as a demonstration that the task considered in Gigastep allows intelligent and collaborative behavior to emerge through MARL algorithms. For instance, the learning curves and win rates from the previous section could have been obtained by agents that learn to exploit issues in the environment definition instead of learning genuine behaviors. Our analysis concludes that this is not the case, and Gigastep is a suitable MARL benchmark allowing the learning of collaborative agents.

We study the identical 20 vs 20 and the special 5 vs 1 scenarios here. A diverse set of behaviors was discovered using the baseline training method. For instance, in Figure 10 we observed a line formation emerging. Figure 11 shows the red team encircling an agent from the opposite team. In Figure 12 we observe that the agents have learned to climb and build up potential energy at the beginning of the episode. Finally, Figure 13 shows that a group of agents has learned to stay together.

## 3 Training Hyperparameters

The hyperparameters of the baseline MARL training using MA-PPO are shown in Table 1.

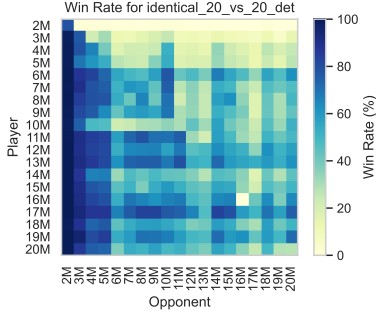

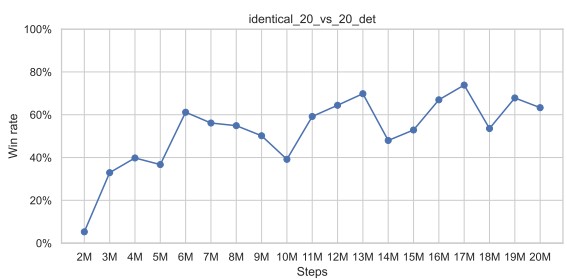

(a) Win rate when playing each policy check-point against each other checkpoint.

(b) Accumulated win rate of the different checkpoints (x-axis) with respect to all other checkpoints.

Figure 3: Learning progress of self-play with multi-agent PPO on the `identical_20_vs_20` scenario.

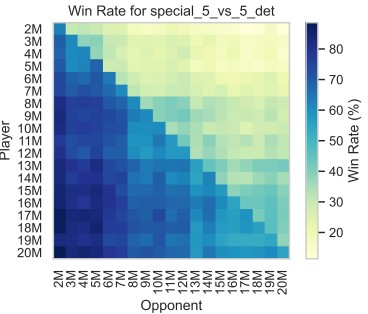

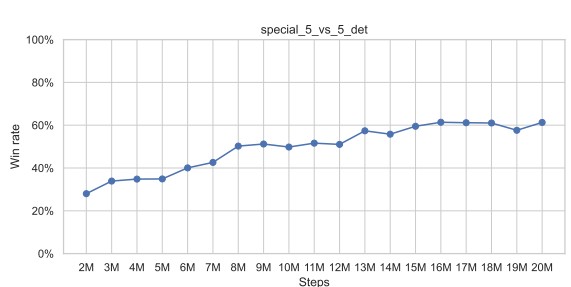

(a) Win rate when playing each policy check-point against each other checkpoint.

(b) Accumulated win rate of the different checkpoints (x-axis) with respect to all other checkpoints.

Figure 4: Learning progress of self-play with multi-agent PPO on the `special_5_vs_5` scenario.

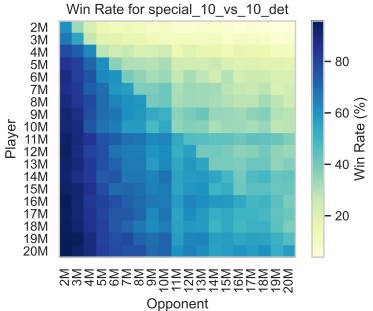

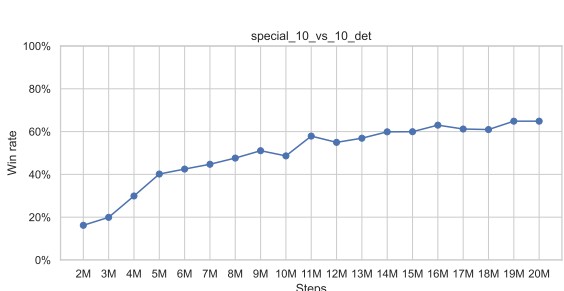

(a) Win rate when playing each policy check-point against each other checkpoint.

(b) Accumulated win rate of the different checkpoints (x-axis) with respect to all other checkpoints.

Figure 5: Learning progress of self-play with multi-agent PPO on the `special_10_vs_10` scenario.

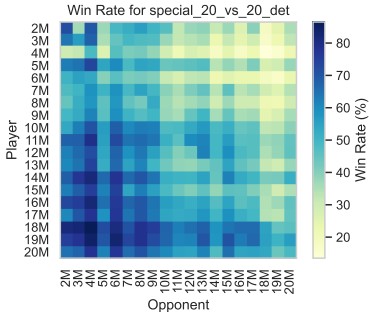

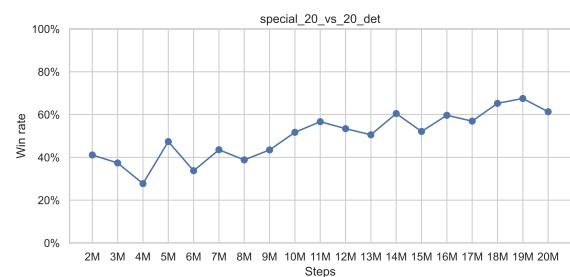

(a) Win rate when playing each policy check-point against each other checkpoint.

(b) Accumulated win rate of the different checkpoints (x-axis) with respect to all other checkpoints.

Figure 6: Learning progress of self-play with multi-agent PPO on the `special_20_vs_20` scenario.

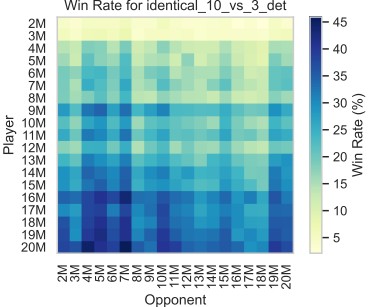

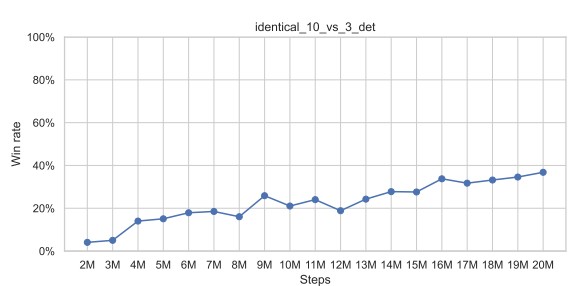

(a) Win rate when playing each policy check-point against each other checkpoint.

(b) Accumulated win rate of the different checkpoints (x-axis) with respect to all other checkpoints.

Figure 7: Learning progress of self-play with multi-agent PPO on the `identical_10_vs_3` scenario.

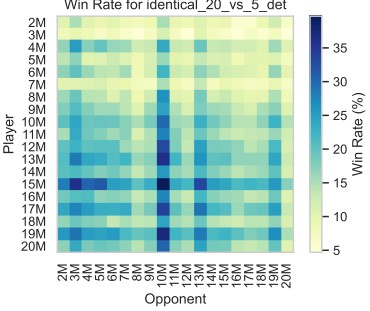

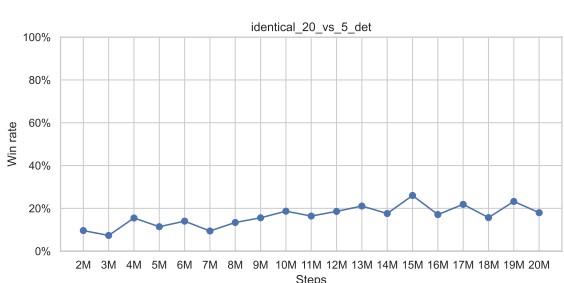

(a) Win rate when playing each policy check-point against each other checkpoint.

(b) Accumulated win rate of the different checkpoints (x-axis) with respect to all other checkpoints.

Figure 8: Learning progress of self-play with multi-agent PPO on the `identical_20_vs_5` scenario.

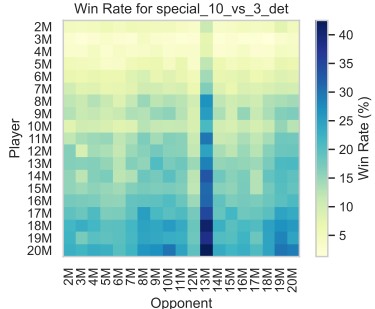
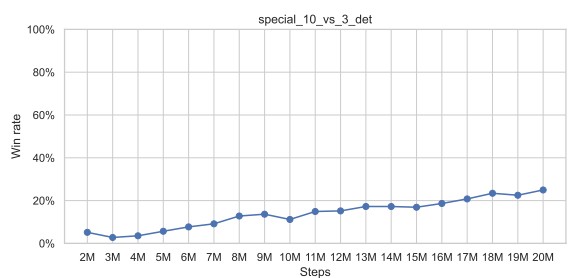

(a) Win rate when playing each policy check-point against each other checkpoint.

(b) Accumulated win rate of the different checkpoints (x-axis) with respect to all other checkpoints.

Figure 9: Learning progress of self-play with multi-agent PPO on the `special_10_vs_3` scenario.

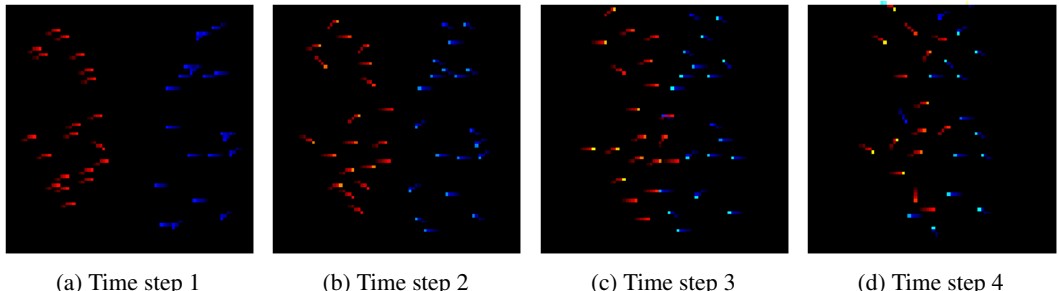

(a) Time step 1      (b) Time step 2      (c) Time step 3      (d) Time step 4

Figure 10: Line formation (red agents flying behind each other).

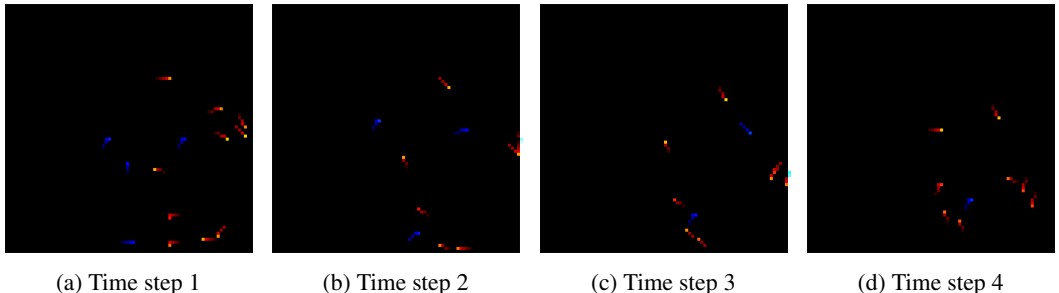

(a) Time step 1      (b) Time step 2      (c) Time step 3      (d) Time step 4

Figure 11: A group of red agents encircling an adversary agent.

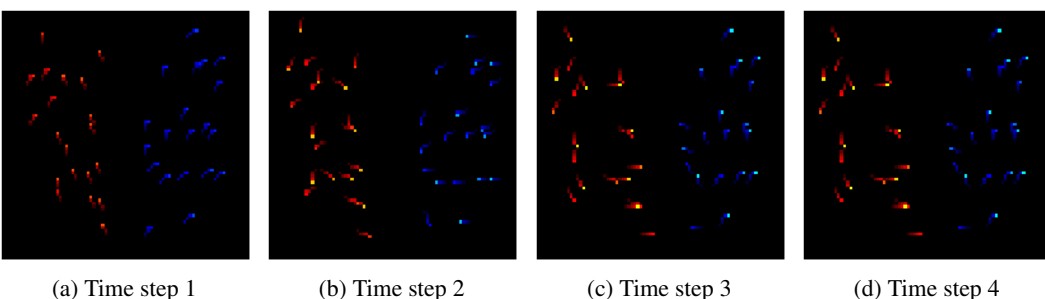

(a) Time step 1      (b) Time step 2      (c) Time step 3      (d) Time step 4

Figure 12: Synchronized climbing (the color indicates the altitude of an agent)

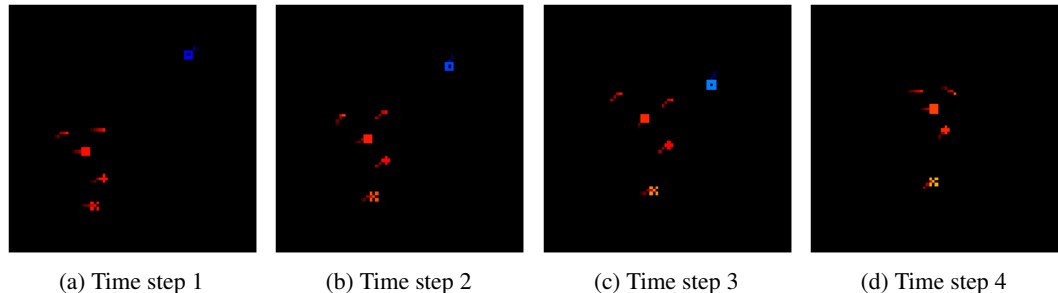

| (a) Time step 1 | (b) Time step 2 | (c) Time step 3 | (d) Time step 4 |

Figure 13: Chasing of an adversary as a group.

Table 1: Hyperparameters used in the training of the MA-PPO baseline agents

| Parameter | Value |
| --- | --- |
| lr | 1e-3 |
| value loss coef | 0.5 |
| batch size | 1000 |
| num steps | 8 |
| num mini batch | 4 |
| use linear lr decay | True |
| entropy coef | 0.02 |
| clip param | 0.2 |
| gae gamma | 0.95 |
| max grad norm | 0.1 |

## 4 Evolutionary Multi-agent RL

In this section, we provide some baselines for training Gigastep agents using evolutionary algorithms. This part serves as a demonstration to show how gradient-free methods can be used in conjunction with Gigastep. Figure 14 provides win-rates as well as non-lose-rates against the baseline policy pool of agents trained via Open-ES, SNES, and DES on the *Identical 5 vs. 5* task over 2000 generations. It can be observed that the resulting agents continuously improve over the course of training and display low lose-rates against the baseline policy pool, underlining the promise of combining Gigastep environments with multi-agent adaptations of evolutionary search algorithms.

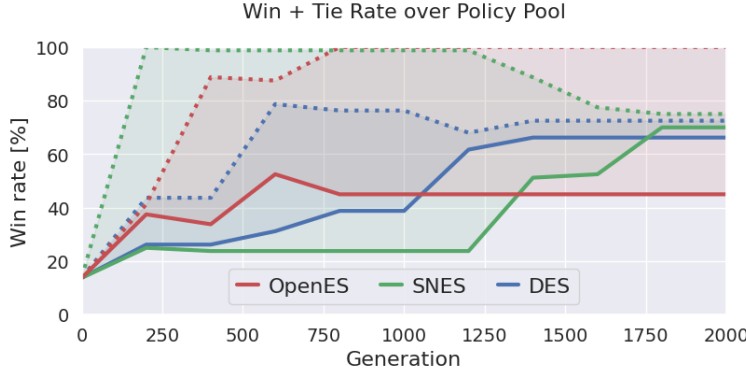

Figure 14: Evaluation of evolutionary algorithms Open-ES, SNES, and DES on Gigastep scenario *Identical 5 vs. 5*. Both the win-rate (solid) and win-or-tie-rate (dotted) improve over the course of training to outperform the baseline policy pool.

Scanario name template: $\mathrm{B\_N\_vs\_M\_U\_X\_Y\_Z}$

      B: Task (B $\in$ {`waypoint, hide_and_seek, identical, special`})
      N: Number of agent in team A (N$\geq$ 1)
      M: Number of agent in team B (M$\geq$ 1)
      U: Full or partial observability (U $\in$ {`fobs, pobs`})
      X: Observation type (X $\in$ {`vec, rgb`})
      Y: With or without obstacles (Y $\in$ {`void, maps`})
      Z: Action space (Z $\in$ {`disc, cont`})

Examples
      `identical_20_vs_5_fobs_rgb_maps_disc`
      `waypoint_5_vs_5_pobs_vec_void_cont`
      ⋮

Figure 15: Naming convention of the built-in scenarios

## 5 Built-in Scenarios

The result of a MARL training process with Gigastep depends on the setting of the scenario. For instance, the fully observable mode arguably allows a much more stable and faster training than the partially observable mode due to the inherent stochasticity of the agent observations. Moreover, RGB image observations introduce another dimension of difficulty to the task as agent information such as locations, velocity, etc. need to be extracted from a high-dimensional observation space compared to the feature vector observations.

To allow a fair comparison between MARL algorithms, it is essential that these scenario settings are kept constant. Consequently, to specify the settings, we define a naming scheme for Gigastep's built-in scenarios. The scheme is shown in Figure 15. Concretely, the naming starts with the objective, followed by the specification of the number of agents each time. The scenario name is then followed by markers indicating the observability mode (fully or partially observable), the observation type (RGB or vector), whether the map contains obstacles (void or maps), and finally concluded by the specification of discrete or continuous action space.

Gigastep allows building custom scenarios containing different objectives, agent properties, and team compositions with minimal code. We expect and support the formulation of additional scenarios by the MARL research community, and will include them in future updates of Gigastep.