# OpenReview forum: "Gigastep - One Billion Steps per Second Multi-agent Reinforcement Learning"
_NeurIPS.cc/2023/Track/Datasets_and_Benchmarks — NeurIPS 2023 Datasets and Benchmarks Poster_

### Official Review · Reviewer_rWA8 · 2023-07-07
**Review of GigaStep**

**Rating:** 3
**Confidence:** 4
**Correctness:** Nothing obviously incorrect.

**Strengths:**

- I agree with the exposition - a better many-agent MARL environment is needed. Existing ones are either too trivially simple, or too compute-heavy / difficult to work with. SMAC, the most popular similar environment to this work, seems to be limited to only running one copy of the environment at a time, and not very quickly, so it would be very hard to do useful research with.
- I appreciated the table of related work
- this work seems loosely modeled off of the dynamics of SMAC, which seems to be popular despite how difficult it is to work with. i think there's room to make a "SMAC knockoff but easier to work with"

**Additional Feedback:**

I would need to be confident that the environment requires non-trivial agent intelligence and has a solid metric for comparing progress against to consider recommend accepting this paper. But I do think the problem is clear and important, and if developed further, that this work could be a useful contribution to the field.

**Clarity:**

The paper has many small writing mistakes, it needs a close editing. There's other clarity improvements that could be made but I'll focus on those if the fundamentals of the paper get resolved.

**Documentation:**

Yes

**Ethics:**

No concerns.

**Limitations:**

- Calling it a "3d" environment when the graphics are 2d is misleading to a reader
- The action space and dynamics are extremely simple. Seems like the entirety of the dynamics are "agents can move, they can collide with parts of the map or other agents, and reward can be given if agents get to certain locations". Whereas in SMAC, agents can shoot, there are different types of agents with different capabilities, agents have health, etc.
- Is there even the potential for interesting emergent strategies given these simple dynamics?
    - eg in SMAC, the authors provided some examples of strategies you'd expect to see emerge (given knowledge of human gameplay). I have a hard time imagining that much intelligence is required to solve these very simple challenges presented in this environment. As a researcher, I don't feel convinced that this provides a real challenge to my agents.
- Relying on self-play seems bad for a benchmark. How do future researchers using this environment compare their results to others? I don't think ELO is useful/stable as a comparative metric? Your results vs the hand-coded agents are already close to 100% win rate, so those aren't useful for future research.
- I think many of the best benchmarks present a single clear metric that starts out not-great, which future research can seek to improve. The value of the benchmark is essentially presenting an open problem to future research, and saying "here's how you can clearly tell if you succeeded". This environment doesn't clearly do that.

- I would like to see video of the environment and of the trained strategies - and analysis of the strategies. It's hard to get a feel for an environment just from descriptions and a couple still images.


**Opportunities For Improvement:**

(please read limitations section first, will build on that)

- I would make the dynamics more complex. Probably the simplest approach is to try to copy SMAC closely - add shooting and health and different agent types and stuff. Copying a game that you know has interesting dynamics (and that people are familiar with) ensure that it will be useful and understandable.
- You need a benchmark metric that isn't self-play. Perhaps make stronger hand-coded AIs, perhaps make imbalanced scenarios, perhaps make agent-vs-environment scenarios. It needs to be clear that there's a good bit of future research required to solve the benchmark (ie it's not already solved), but also it needs to be clear that the benchmark is solveable / improveable - you can't just set up impossible scenarios and say "well the win rate is low, there must be room for improvement". To get someone to work on your benchmark, they need to be confident that both of these properties are true.
- Don't worry so much about getting it absurdly fast - you don't need anywhere near a billion steps per second to do good research. Existing envs do like 100 steps/sec and are still used. As long as you can do say 10k steps/sec (in total across parallel environments) on a machine it's plenty fast.

**Relation To Prior Work:**

The clarity could be improved some, but it's relatively clear.

**Summary And Contributions:**

Gigastep is a multi-agent RL environment focused on providing a useful environment for many-agent RL research without requiring large amounts of compute. Most existing multi-agent RL environments are either focused on a very small number of agents, are computationally intensive and hard to work with (eg SMAC), or have trivially simple dynamics.

The action space is limited to enabling motion of the agent in 3 dimensions. The observation space can either be a state vector or a 2d top-down view of the agents, where z-position reflected using colors. The observation space can be partially observable, where the agents only see nearby other agents. The reward can be configured in different scenarios, so far including tagging, moving to a waypoint, and hide-and-seek.

The environment is simulated on the GPU and thus can run quickly both with a single environment and with many environments in parallel.

The authors evaluate the environment using some existing MARL algorithms against hand-coded agents and in self-play.

---

### Official Review · Reviewer_PUtM · 2023-07-20
**Brilliant contribution, will be very useful to MARL researchers**

**Rating:** 9
**Confidence:** 4
**Correctness:** Seems correct and the things are clea…
**Clarity:** Clear

**Strengths:**

A very comprehensive literature survey, Table. 1 clearly places the contributions of this work. I like this way of doing things.

The implementation in JAX and specifically leveraging XLA leads to quite a substantial improvement in compute time.

Works on both continuous and discrete action spaces.

Works with vectors and images

Very flexible and is useful for a large number of applications.


It seem like any typical RL algorithm and corresponding test application can be programmed with this framework. Is it true?

**Additional Feedback:**

This is a brilliant paper and has a lot of uses to the RL community. I think the paper is very nicely written.

**Documentation:**

Yeas, the readme in the supplementary files is detailed.

**Limitations:**

The limitations are appropriately discussed.

**Opportunities For Improvement:**

One thing that is not clear to me is how does this framework work with different environments. Some discussion should be made regarding that.

Does this environment also  translate to single agent environments? For instance, how would one go about thinking common environments such as atari games in their context.





**Relation To Prior Work:**

Great related works section. Table 1 explains it properly.

**Summary And Contributions:**

Very valid and interesting problem. RL training is some of the slowest process we can go through even for a single agent, that is.
I think, the motivation for this benchmark is ideal and it would have real use to everyone. Therefore the contributions provided by the Gigastep benchmark are significant.

---

### Official Review · Reviewer_6dKh · 2023-07-22
**A Multi-Agent Benchmark for Decentralized Partially Observable Markov Decision Processes**

**Rating:** 7
**Confidence:** 3
**Correctness:** The evaluation methods are decent MAR…
**Clarity:** The paper is written quite well.

**Strengths:**

- The benchmark paper addresses the challenging and relevant problem of decentralized partially observable Markov decision processes (Dec-POMDPs) in multi-agent environments. This problem is important in various fields, including artificial intelligence, reinforcement learning, and multi-agent systems.
- Gigastep is designed to be computationally accessible, making it suitable for a wide range of researchers and institutions, including those with limited resources.
- By providing a benchmark that tackles a challenging problem and offering accessible code, Gigastep has the potential to impact the research community significantly. It can serve as a standard evaluation platform for developing and comparing new multi-agent algorithms and strategies.
- The benchmark paper is well-structured and effectively presents the key design considerations. The explanations are clear and concise, making the material more approachable for readers with varying levels of expertise.

**Additional Feedback:**

1. Do you have any plan for surporting the benchmark for decentralized training?
2. In order to achieve a processing speed of one million steps per second, a batch size of 8,000 is utilized. My question is, whether the stated one million steps represent the total steps taken collectively by all agents or just the steps taken by an individual agent. Additionally, whether the chosen batch size of 8,000 is reasonable for a typical training process?

**Documentation:**

There are sufficient details on the design of the Gigastep benchmark.

**Ethics:**

No.

**Limitations:**

The authors have addressed the limitations in a separate section.

**Opportunities For Improvement:**

1. While there is already some test code, it falls short of providing comprehensive unit tests for the environments in Gigastep. Unit tests are crucial for ensuring the correctness and reliability of the codebase. Without them, it may be challenging for other researchers to verify the integrity of the implemented environments, potentially leading to discrepancies in results and hindering reproducibility.

2. The current codebase lacks detailed documentation, making it challenging for other researchers to understand and extend the benchmark.

3. The paper does not adequately address a detailed maintenance plan for the Gigastep codebase. A well-defined maintenance plan would enhance the benchmark's longevity and usefulness, providing stability and support for future researchers.

**Relation To Prior Work:**

I appreciate the comprehensive comparison with the previous MARL benchmark environments in Section 2 and Table 1.

**Summary And Contributions:**

The challenge in decentralized partially observable Markov decision processes lies in dealing with multiple agents operating with private information while needing to reason about the global state of the environment. This work aims to address this challenge by introducing Gigastep, a multi-agent environment that is designed to be both computationally accessible to underfunded institutions and sufficiently complex. The key features of Gigastep are as follows:

1. Gigastep is implemented entirely in JAX, providing native vectorization support, enabling JIT compilation for the agents' states and actions.
2. The environment includes a variety of agent roles, such as collaborators and adversaries, fostering the development of adaptive strategies to cope with an ever-changing environment.
3. Uncertainty is incorporated into the environments for additional complexity.

---

### Official Review · Reviewer_zXRT · 2023-07-22
**Interesting MARL environment**

**Rating:** 7
**Confidence:** 4
**Correctness:** The submission is an RL environment w…
**Clarity:** The paper is well-written and easy to…

**Strengths:**

+ Existing landscape of Dec-POMDP either has very high complexity and requires special hardware beyond the reach of most in the community, or has very low complexity environments that fail to present the intricacies of the real world scenarios. Gigastep addresses this challenge.

+ Gigastep can be run on commodity-grade hardware,

+ It supports both collaborative and adversarial components.

+ Dynamics incorporated stochasticity.



**Additional Feedback:**

Please see suggestions for improvement.

**Documentation:**

The RL environment definition is well-documented

**Ethics:**

There are no ethical concerns.

**Limitations:**

There is no concern of negative societal impact.Th

**Opportunities For Improvement:**

Plots in Figure 4 can use larger fonts.



**Relation To Prior Work:**

Related work is well-discussed.

**Summary And Contributions:**

The paper presents Gigastep a fully vectorizable multi-agent reinforcement learning (MARL) environment implemented in JAX and capable of a billion environment steps on a consumer-grade hardware. The environment supports both continuous and discrete action space, provides RGB image and feature vector observations and also supports both collaborative and adversarial tasks. Extensive experiments are conducted to validate Gigasteps.

---

### Decision · Program_Chairs · 2023-09-22

**Decision:**

Accept (Poster)

**Comment:**

The paper presents a new environment for MARL on decentralised POMDPs, designed in JAX for extreme speed.  This environment consists of ‘airborne agents’ navigating 3d space in teams, though rendered in 2d using colour for height, with different types of objectives’ such as tagging, chasing way points & hide and seek.

**Perceived Strengths:**
* Fast to simulate. Since the environment can be stepped on GPU, it can be simulated very quickly, potentially up to a billion steps a second.  This is useful for tabula rasa RL which is notoriously sample inefficient, and where CPU environments normally are the bottlenecks.
* The environment is stochastic, with collaborative and adversarial dynamics, which adds to the complexity of the dynamics

**Perceived Weakness:**
* No clear challenge. Reviewer rWA8 suggests that an environment with such a simple action space and dynamics may not require non-trivial agent intelligence to solve.  Authors responded that the agents in the game are heterogenous, and also draw a compelling parallel between the shooting dynamics of smac and their tagging dynamic.
* No clear benchmark metric. Reviewer rWA8 suggests that no clear metric is presented indicate ‘skilled’ or ‘top-level’ game play, which would indicate when the game is ‘solved’. They point out that self-play by itself is not a good metric, and the authors agree and agree to add new pretrained against to the benchmark.
* Lack of documentation. While some praise the documentation (presumably on how to run the env), there is a complaint that it is difficult to know how to extend the benchmark environment - something that would be important in such a simple environment.


**AC View:**

This paper attracted some very mixed reviews.

Reviewer PUtM rated this paper a 9, in the top 15% of papers, with a short review highlighting the comprehensive literature survey, fast simulation speed, and flexibility with respect to different action spaces, and observation spaces.  I found this review insufficient to recommend such a high score and discounted this score somewhat.

Reviewer rWA8 rated this paper a 3, a clear rejection, pointing out that, while fast environments are very useful, the key purpose of an environment is to pose a _challenge_ to learning methods, and allow one to _measure progress_ in this task.  In this criticism I believe they rightly get to the heart of the matter of what was missing in the paper - which was a focus on the environment, the challenge and the intelligence required to succeed in it.  I found this criticism compelling, and was pleased to see the reviewers engaging with it, highlighting the challenge of their environment and adding pretrained agents for benchmarking, and demonstrating complex behaviours in the appendix. Regrettably, this reviewer did not engage with the author's responses, undermining their argument somewhat.

Two other reviews of varying length, considered this paper a clear acceptance, albeit with reviews of varying depth.

In all, I recommend narrow acceptance of this paper, noting the extensive responses and the changes made to improve the paper in response to reviewer rWA8 as a decisive factor in my decision.  For example, I felt the highlighting of complex multi agent phenomena, as indicated in the supplementary material,  showed some degree of 'non-trivial intelligence' that rWA8 was looking for.